# Safinamide Improves Non-Motor Symptoms Burden in Parkinson’s Disease: An Open-Label Prospective Study

**DOI:** 10.3390/brainsci11030316

**Published:** 2021-03-02

**Authors:** Diego Santos García, Carmen Labandeira Guerra, Rosa Yáñez Baña, Maria Icíar Cimas Hernando, Iria Cabo López, Jose Manuel Paz Gonález, Maria Gemma Alonso Losada, María José González Palmás, Cristina Martínez Miró

**Affiliations:** 1Department of Neurology, CHUAC, Complejo Hospitalario Universitario de A Coruña, 15006 A Coruña, Spain; Jose.Maria.Paz.Gonzalez@sergas.es (J.M.P.G.); Cristina.Martinez.Miro@sergas.es (C.M.M.); 2Department of Neurology, CHUVI, Complejo Hospitalario Universitario de Vigo, 36213 Vigo, Spain; carmen.labandeira@hotmail.com (C.L.G.); gemavarita@gmail.com (M.G.A.L.); 3Department of Neurology, CHUO, Complejo Hospitalario Universitario de Ourense, 32005 Ourense, Spain; Rosa.Yanez.Bana@sergas.es; 4Department of Neurology, Hospital de Povisa, 36211 Vigo, Spain; icimash@povisa.es; 5Department of Neurology, CHOP, Complejo Hospitalario Universitario de Pontevedra, 36002 Pontevedra, Spain; icabol@yahoo.es (I.C.L.); Maria.Josefa.Gonzalez.Palmas@sergas.es (M.J.G.P.)

**Keywords:** effectiveness, non-motor symptoms, open-label study, Parkinson’s disease, safinamide

## Abstract

Some studies observed a benefit of Parkinson’s disease (PD) patients after treatment with safinamide in some non-motor symptoms (NMSs). The aim of this study was to analyze the effectiveness of safinamide on NMS burden in PD. SAFINONMOTOR (an open-label study of the effectiveness of safinamide on non-motor symptoms in Parkinson’s disease patients) is a prospective open-label single-arm study conducted in five centers from Spain. The primary efficacy outcome was the change from baseline (V1) to the end of the observational period (6 months) (V4) in the non-motor symptoms scale (NMSS) total score. Between May/2019 and February/2020 50 patients were included (age 68.5 ± 9.12 years; 58% females; 6.4 ± 5.1 years from diagnosis). At 6 months, 44 patients completed the follow-up (88%). The NMSS total score was reduced by 38.5% (from 97.5 ± 43.7 in V1 to 59.9 ± 35.5 in V4; *p* < 0.0001). By domains, improvement was observed in sleep/fatigue (−35.8%; *p* = 0.002), mood/apathy (−57.9%; *p* < 0.0001), attention/memory (−23.9%; *p* = 0.026), gastrointestinal symptoms (−33%; *p* = 0.010), urinary symptoms (−28.3%; *p* = 0.003), and pain/miscellaneous (−43%; *p* < 0.0001). Quality of life (QoL) also improved with a 29.4% reduction in the PDQ-39SI (from 30.1 ± 17.6 in V1 to 21.2 ± 13.5 in V4; *p* < 0.0001). A total of 21 adverse events in 16 patients (32%) were reported, 5 of which were severe (not related to safinamide). Dyskinesias and nausea were the most frequent (6%). Safinamide is well tolerated and improves NMS burden and QoL in PD patients with severe or very severe NMS burden at 6 months.

## 1. Introduction

Parkinson’s disease (PD), the second most common neurodegenerative disease after Alzheimer’s disease, is a progressive neurodegenerative disorder causing motor and non-motor symptoms (NMS) that result in disability, loss of patient autonomy and caregiver burden [1]. With respect to pharmacological treatment, levodopa is the gold standard treatment for PD but other medications are useful: dopamine agonists, COMT inhibitors or MAO-B inhibitors [2]. Safinamide is an oral α-aminoamide derivate marketed for the treatment of PD with both dopaminergic properties, namely highly selective and reversible inhibition of MAO-B, and nondopaminergic properties, namely selective sodium channel blockade and calcium channel modulation, with consequent inhibition of excessive glutamate release [3]. In 2014, safinamide was approved in the European Economic Area, as “an add-on therapy” to stable dose of levodopa, alone or in combination with other PD therapies in fluctuating PD patients. In PD trials, addition of safinamide to levodopa has demonstrated to improve both motor scores and duration of “on time” similar to other MAO-B inhibitors, with and added benefit of not increasing troublesome dyskinesia [4,5,6,7,8]. Moreover, safinamide is safe and well tolerated [9]. Furthermore, data from some studies suggest a possible benefit of PD patients after treatment with safinamide in some NMS such as pain, mood and cognition [10,11,12]. The favorable effect of safinamide on some NMS may be explained not only by its dopaminergic effect but also by its inhibitory action on state- and use-dependent sodium channels and abnormal glutamate release [13]. In fact, recent studies demonstrated an improvement on NMS burden as a whole [10], being relevant because NMS burden impacts negatively on PD patients’ quality of life (QoL) [14]. NMS burden, assessed with the non-motor symptoms scale (NMSS), has been considered as the primary efficacy variable in some recent trials [15].

The aim of the present prospective open-label single-arm study (SAFINONMOTOR, an open-label study of the effectiveness of safinamide on non-motor symptoms in Parkinson’s disease patients) was to analyze the effectiveness of safinamide on NMS burden (defined as the NMSS total score) in PD patients. Secondary objectives were to analyze the effectiveness of safinamide on NMS burden of each domain of the NMSS, and also specifically on sleep, mood, pain, health-related QoL and autonomy for activities of daily living (ADL).

## 2. Material and Methods

SAFINONMOTOR is a mono-country (Spain), multicenter, observational (phase IV), prospective, open-label, follow-up study. Five neurology sites from Galicia (Spain) dealing with PD participated. A total of 50 consecutive PD patients were expected to be included in the study, corresponding to 10 patients/site. Inclusion criteria were: (1) diagnosis of Parkinson’s disease according to the United Kingdom Parkinson’s Disease Society Brain Bank criteria [16]; (2) to have the indication of receiving safinamide according to the neurologist criteria; (3) to have a total NMSS score at baseline ≥ 40; (4) no dementia criteria with a MMSE at baseline ≥ 26) [17]; (5) older than 30 years old and (6) to wish to voluntarily participate and to sign a consent form. Exclusion criteria were: (1) to be under MAO-B inhibitor therapy (rasagiline or selegiline); (2) any other contraindication to be treated with safinamide according to product data; (3) incapacity to complete the questionnaires adequately; (4) other disabling concomitant neurological disease (stroke, severe head trauma, neurodegenerative disease, etc.); (5) other severe and disabling concomitant non-neurological disease (oncological, autoimmune, etc.); (6) expected impossibility of long-term follow-up and (7) patient who was participating in a clinical trial and/or other type of study. All the neurologists who participated in the study were experts on PD/movement disorders.

The study visits included: (1) V1 (baseline); (2) V2 (1 months ± 7 days); (3) V3 (3 months ± 15 days); (4) V4 (6 months ± 15 days, end of the observational period). At baseline (V1), after providing written informed consent to participate in the study, subjects completed an assessment that included motor symptoms (Hoehn and Yahr (H&Y) [18]; unified Parkinson’s disease rating scale (UPDRS) part III and part IV [19] and freezing of gait questionnaire (FOGQ) [20]), NMS (NMSS [21]; Epworth sleepiness scale (ESS) [22]; Pittsburgh sleep quality index (PSQI) [23]; Beck depression inventory-II (BDI-II) [24]; King’s Parkinson’s disease pain scale (KPPS) [25]; VAS-PAIN [26] and visual analog fatigue scale (VAFS) [27]), disability (Schwab and England activities of daily living scale (ADLS) [28]) and health related QoL (the 39-item Parkinson’s disease questionnaire (PDQ-39) [29]). The same assessment was performed at V2, V3 and V4 (final visit) except for UPDRS-IV (only at V1 and V4) and H&Y (only at baseline). Moreover, patient global impression of change (PGIC) [30] was conducted at V2, V3 and V4. Information on sociodemographic aspects, factors related to PD, comorbidity and treatment was collected.

The primary objective was to analyze the effectiveness of safinamide on NMS burden (defined as the NMSS total score) at 6 months. Secondary objectives included: (1) to analyze the effectiveness of safinamide on NMS burden of each domain of the NMSS, and specifically also on diurnal somnolence (ESS), sleep (PSQI), mood (BDI-II) and pain (KPPS); (2) to analyze the effectiveness of safinamide on health related QoL (PDQ-39) and functional capacity for ADL (ADLS); (3) to assess the clinical global impression of change according to the patient (PGIC) and (4) to analyze the safety and security of safinamide in PD patients.

Safinamide was administered as a once-daily 50 mg pill for 1 month and switched to 100 mg/day at V2. However, in some cases (e.g., dyskinesia) the dose of 100 mg could be introduced earlier or the dose could be kept at 50 mg/day according to the criteria of the neurologist. Patients could be receiving any other antiparkinsonian drugs: levodopa, dopamine agonist, COMT inhibitor, amantadine and/or anticholinergic. During follow-up, any other medications different of safinamide should not been modified (regimen, doses, etc.) except if the neurologist considered these changes absolutely necessary. All the changes including PD and not-PD related medications and levodopa equivalent daily dose (LEDD) [31] of levodopa was recorded.

### 2.1. Data Analysis

Data were processed using SPSS 20.0 for Windows. Continuous variables were expressed as the mean ± SD or median and quartiles, depending on whether they were normally distributed. Relationships between variables will be evaluated using the Student’s *t*-test, the Mann–Whitney U test, Spearman’s or Pearson’s correlation coefficient as appropriate (distribution for variables was verified by one-sample Kolmogorov–Smirnov test). NMS burden was defined as: mild (NMSS 1–20); moderate (NMSS 21–40); severe (NMSS 41–70) and very severe (NMSS > 70) [32]. Each domain of the NMSS and KPPS was expressed as a percentage: (score/total score) × 100. The PDQ-39 was expressed as a summary index (PDQ-39SI): (score/156) × 100.

The primary efficacy outcome was the change from baseline (V1) to the end of the observational period (6 months) (V4) in the NMSS total score. The change from V1 to V4 in NMSS domains, ESS, PSQI, BDI-II, KPPS, PDQ-39 and ADLS were the secondary efficacy outcome variables. Analyses on efficacy variables were performed with the ITT data set (all subjects who receive at least 1 pill of safinamide and had a baseline and treatment observation for the primary efficacy outcome measure). A paired-sample *t*-test or Wilcoxon’s rank sum test as appropriate was performed for testing the change from baseline. Values of *p* < 0.05 were considered significant.

The safety data set consists of all subjects for whom the study device was initiated. Safety analyses were assessed by adverse events (AEs). All AEs was coded using the current version of the Medical Dictionary for Regulatory Activities (MedDRA). The number and percentage of subjects with treatment emergent AEs by the MedDRA system organ class and preferred term, by severity, and by the relationship to study treatment as assessed by the investigator was provided for overall subjects.

### 2.2. Standard Protocol Approvals, Registrations and Patient Consents

For this study, we received approval from the Comité de Ética de la Investigación Clínica de Galicia from Spain (2018-052; 28/FEB/2019). Written informed consents from all participants in this study were obtained before the start of the study. SAFINONMOTOR was classified by the AEMPS (Agencia Española del Medicamento y Productos Sanitarios) as a post-authorization prospective follow-up study with the code DSG-SAF-2018-01.

### 2.3. Data Availability

The protocol and the statistical analysis plan are available on request. Deidentified participant data are not available for legal and ethical reasons.

## 3. Results

A total of 50 out of 54 PD patients were included between May/2019 and February/2020 (age 68.5 ± 9.12 years; 58% females). Four patients initially selected were excluded for different reasons: three for presenting a NMSS total score < 40 and one due to have already started with safinamide at V1. Data about sociodemographic aspects, comorbidities, antiparkinsonian drugs and other therapies are shown in Table 1. The mean time from diagnosis of PD was 6.4 ± 5.1 years. All patients except three were receiving levodopa, two patients were under levodopa/carbidopa infusion therapy and none were with apomorphine or deep brain stimulation. The mean LEDD was 810.2 ± 518.1 (range from 100 to 2.350 mg).

At baseline (V1), 78% (39/50) of the patients presented with motor fluctuations and 30% (15/50) with dyskinesia. The mean UPDRS-III during the ON state was 24.6 ± 9.1. With regards to the NMS, the mean NMSS total score at baseline was 97.5 ± 43.7, presenting 68% (34/50) of the patients with very severe NMS burden (NMSS total score > 70). Considering the different domains from the NMSS, the highest scores were in domains 7 (urinary symptoms), 2 (sleep/fatigue) and 3 (mood/apathy) (Table 2). With regard to QoL, the most affected domains were 8 (bodily discomfort), 3 (emotional well-being) and 1 (mobility) (Table 2).

At 6 months, 44 patients completed the follow-up (88%). The NMSS total score was reduced by 38.5% (from 97.5 ± 43.7 in V1 to 59.9 ± 35.5 in V4; *p* < 0.0001) (Table 2). Eight patients presented a higher NMSS total score at the end of the follow-up compared to baseline (18%) whereas in the rest of the patients the NMSS total score was lower (range from 3 to 105 points). In four patients who improved, the NMSS total score was reduced in ≤10 points. Compared to the score at V1, the change at V2 and V3 was significant too (*p* < 0.0001 for both analysis) (Figure 1 and Appendix A). By domains, improvement was observed in sleep/fatigue (−35.8%; *p* = 0.002), mood/apathy (−57.9%; *p* < 0.0001), attention/memory (−23.9%; *p* = 0.026), gastrointestinal symptoms (−33%; *p* = 0.010), urinary symptoms (−28.3%; *p* = 0.003) and pain/miscellaneous (−43%; *p* < 0.0001) (Table 2 and Figure 2). At the end of the follow-up, 34.1% of the patients presented with severe NMS burden and another 34.1% with very severe NMS burden, but 31.8% with a slight and/or moderate NMS burden (NMSS 1–40) (Figure 3). It was observed a significant reduction in the score of other scales used for the assessment of motor and NMS: UPDRS-III during the ON state (−17.9%; *p* = 0.009); ESS (−24.7%; *p* = 0.012); PSQI (−19.8%; *p* = 0.001); BDI-II (−35.9%; *p* < 0.0001) and KPPS (−43.6%; *p* < 0.0001).

QoL also improved at V4 with a 29.4% reduction in the PDQ-39SI (from 30.1 ± 17.6 in V1 to 21.2 ± 13.5 in V4; *p* < 0.0001) compared to the score at baseline. Specifically by domains, the difference between V1 and V4 was significant for PDQ-39SI-1 (Mobility) (*p* = 0.037), PDQ-39SI-2, (Activities of daily living) (*p* = 0.014), PDQ-39SI-3 (Emotional well-being) (*p* < 0.0001), PDQ-39SI-4 (Stigmatization) (*p* = 0.021) and PDQ-39SI-8 (Pain and discomfort) (*p* = 0.018). At 6 months, 28 patients out of 44 (63.7%) felt better regarding the PGIC: 6 very much improved; 12 much improved; 10 minimally improved; 11 no changes and 5 minimally worse.

A total of 21 adverse events in 16 patients (32%) were reported, 5 of which were severe (not related to safinamide) (Table 3). Dyskinesias and nausea were the most frequent (6%). The reasons for withdrawing from the study of the 6 patients were: 1 withdrawal of consent; 1 discontinuation of safinamide after deep brain stimulation procedure (it was recorded as SAE due to hospitalization process); 1 personal decision due to no effect and 3 due to an adverse event (2 dizziness; 1 respiratory infection). Only one patient discontinued due to an adverse event related to safinamide (dizziness). All patients were receiving safinamide 50 mg/day at V2 except 3 cases who were receiving 100 mg/day whereas all patients were receiving 100 mg/day at V3 and V4 except 1 and 2 cases, respectively, who were receiving 50 mg/day. Only 3 patients were receiving rasagiline, which was withdrawn with a washout period of at least 2 weeks before starting safinamide.

## 4. Discussion

The present study observed that NMS burden as a whole improved in PD patients 6 months after starting with safinamide. Specifically, mood (NMSS-domain 3 and BDI-II), sleep (NMSS-domain 2 and PSQI) and diurnal somnolence (ESS), pain (NMSS-domain 9 and KPPS), cognition (NMSS-domain 5) and gastrointestinal (NMSS-domain 6) and urinary symptoms (NMSS-domain 7) improved. Moreover, the effect was significant with 50 mg/day at the first month after starting with safinamide and an improvement in QoL was observed as well. Although other studies have demonstrated an improvement in NMS with safinamide [10,11,12,33,34], this is the first prospective study specifically designed for assessing the change in NMS burden in PD patients after been treated with this drug.

In this study, PD patients had to present with severe or very severe NMS burden (NMSS total score > 40) for being included, but the indication of starting with safinamide was according to the neurologist opinion in clinical practice. It explains that three patients were not receiving levodopa at baseline and the fact that not all the patients presented with motor fluctuations. However, a severe NMS burden can be present in early stages of the disease in PD patients even without motor fluctuations and very importantly, it impacts negatively on the patient’s QoL [35]. The data of the SAFINONMOTOR study at baseline about motor and NMS are in line with other studies [36]. Previously, Bianchi et al. demonstrated a 38.6% reduction in the NMSS total score [10], similar as we observed (38.5%). However, in that study the reduction was of 17.1 points but in ours it was of 27.6 points. This is because in our study the patients’ NMS burden at baseline was severe or very severe and the mean NMSS total score was very high (97.5 points). Furthermore, unlike the Bianchi study, our study was prospective and the change in the NMSS total score was the primary efficacy outcome. Interestingly, we observed a benefit of the patients not only in the NMS burden as a whole but in some NMS too. When different domains of the NMS were considered, an improvement was observed as in Bianchi study in sleep/fatigue, mood/apathy, attention/memory and urinary symptoms. However, and improvement in cardiovascular symptoms and sexual dysfunction was not observed but in gastrointestinal symptoms and miscellaneous was. The most relevant improvement in our study was in mood, with a 57.9% reduction in the NMSS-domain 3 and 35.9% in the BDI-II score. Previously, it was reported that safinamide, compared to placebo, significantly improved the PDQ-39 "Emotional well-being" domain after 6 months and 2 years, and the GRID Hamilton rating scale for depression [37]. In our study, emotional well-being was also the PDQ-39 domain that significantly most improved. Other studies have reported improvement in cognition [34], pain [38,39,40], sleep [41], restless legs syndrome [42] and urinary symptoms [33], in accordance with what we observed. Very recently, Geroin et al. [40] observed after 12 weeks of add-on safinamide therapy, a significant improvement in the primary (KPPS, BPI intensity and interference and NRS) outcomes in 13 PD patients with pain. They used as we, the KPPS, and the results were practically identical than in our study, with a reduction of 19.3 points. However, they did not provide information about the KPPS domains. In PD patients from the SAFINONMOTOR study, a significant improvement was observed in some types of pain typically related to dopamine such as musculoskeletal pain, fluctuation-related pain and nocturnal pain [43]. There is not a clear explanation of what could be the reason for the benefit of PD patients in the different NMS after receiving safinamide. Glutamate has been linked to pain, mood or sleep, and the glutamate or even GABA modulatory effect of safinamide could play a role [13,44]. However, it is well known that many NMS are related to motor fluctuations and to dopamine deficiency and the fact that these symptoms improved with only 50 mg/day of safinamide as well suggests a role, at least in part, of its dopaminergic effect (MAO-B inhibitor). Related to this, the LEDD of safinamide has been recently proposed, being the same for 50 mg than 100 mg (100 mg) [31]. Given that both rasagiline and safinamide are MAOB-Is, one possibility would be to directly compare safinamide versus rasagiline, in such a way that observing benefits with safinamide but not with rasagiline would suggest that it could be at least partly due to an additional action mechanism. For example, a recent clinical observation comparing safinamide and rasagiline as adjunctive therapy in fluctuating PD patients seems to suggest that safinamide, differently from the other MAOB-Is, may improve subjective sleep and daytime sleepiness in motor fluctuating PD [41]. Furthermore, It is also not clear whether safinamide could have a per se antidepressant effect and this influence the perception of other symptoms or vice versa, improve other symptoms and that this also contributes to improving the emotional well-being patient’s perception. Although MAOB-Is may help reduce the severity of depressive symptoms in PD, the positive effect could be only significant in patients with early stage but not in those with middle-to-late-stage PD [45].

Another important observation of the SAFINONMOTOR study is the improvement in QoL. We used the PDQ-39 such as in other studies [46,47]. Cattaneo et al. observed in a recent post hoc analysis using data from the study 018 a reduction in the PDQ-39Si score of 4.07 points [47]. In our study it was of 8.83 points, but again as in the case of the NMSS, the score at baseline was higher. Moreover, in our study we conducted an analysis about the changes in different domains of the PDQ-39 but this has not been properly assessed in many studies [46,47,48]. Not only pain and discomfort and emotional well-being improved but mobility and activities of daily living too, in part probably with regards to the positive effect on the motor status during the ON state, with a reduction in line with other studies in the UPDRS-III score of about 4 points [5,46,47,48]. Borgohain et al. [5] had already observed an improvement in the same domains of the PDQ-39 compared to placebo as in ours except in mobility and activities of daily living. By the contrary, it was not observed in our study a significant improvement in the ADLS score. Finally, although some studies observed an improvement in the freezing of gait with rasagiline [49], we did not observed it with safinamide. The possibility of improving the FOG with MAOB-Is is interesting but, in any case, a study specifically designed to evaluate this would be necessary to obtain conclusions regarding safinamide.

Safinamide was not only effective but also safe and well-tolerated. The results about adverse events are in line with other studies [5,6,7,8,9,10,37,38,39,40,41,46,47,48,50,51,52,53,54]. With regards to this point, a recent study conducted in 1610 PD patients demonstrated a good safety profile of safinamide even in special groups [53]. Dyskinesia, as in some studies, was the most frequent adverse event in our study. Despite this, the frequency was low and the percentage of patients who stopped the treatment was low as well.

Our study has some important limitations. The most important one is related to the study design itself and since there is not a comparative arm with placebo, the results should be interpreted with caution. Moreover and very importantly, the effect of safinamide on NMS was analyzed in PD patients with a severe or very severe NMS burden (NMSS total score > 40), therefore the results cannot be extrapolated to patients with a mild or moderate NMS burden (NMSS total score ≤ 40). For some variables, the information was not collected in all cases. The results were based on scales that collect the opinion of the patient and a bias due to the placebo effect cannot be ruled out. In fact, the improvement in mood could influence the perception of symptoms and the response in other scales [14]. The effect that confinement due to COVID-19 [55] could have on the last months of the follow-up in some of the patients is unknown. Of all the visits conducted, four (at V4) were by telephone due to the pandemic. On the other hand, this is the first study designed to assess the effect of safinamide on NMS burden in PD patients and the first one in which changes in some NMS such as pain, mood or sleep have been exhaustively analyzed. Specific analyses about the observed changes in other variables will be analyzed in the near future.

## 5. Conclusions

In conclusion, safinamide is well tolerated and improves NMS burden and QoL in PD patients. Well-designed studies are necessary to analyze in detail the possible beneficial effect of safinamide on NMS in patients with PD and the mechanisms involved.

## Figures and Tables

**Figure 1 brainsci-11-00316-f001:**
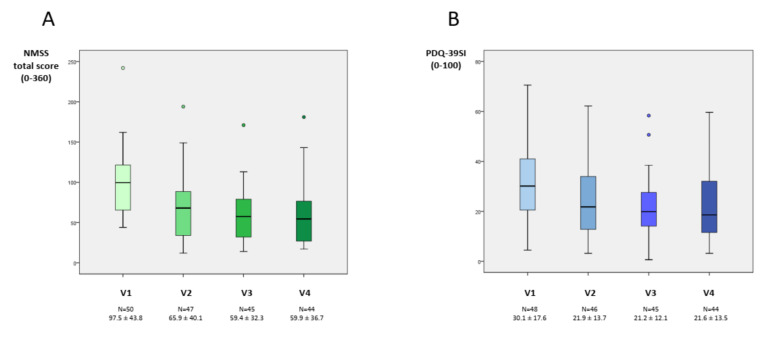
(**A**) NMSS total score at V1 (baseline), V2 (1 months ± 7 days), V3 (3 months ± 15 days) and V4 (6 months ± 15 days). Compared to the score at V1, the change at V2, V3 and V4 was significant (*p* < 0.0001 for all analysis; V2 vs. V1; V3 vs. V1; V4 vs. V1). (**B**) PDQ-39SI at V1, V2, V3 and V4. Compared to the score at V1, the change at V2, V3 and V4 was significant (*p* < 0.0001 for all analysis; V2 vs. V1; V3 vs. V1; V4 vs. V1). Data are presented as box plots, with the box representing the median and the two middle quartiles (25–75%). *p* values were computed using the Wilcoxon signed-rank test. Mild outliers (O) are data points that are more extreme than Q1–1.5 × IQR or Q3 + 1.5 × IQR. NMSs, Non-motor symptoms.

**Figure 2 brainsci-11-00316-f002:**
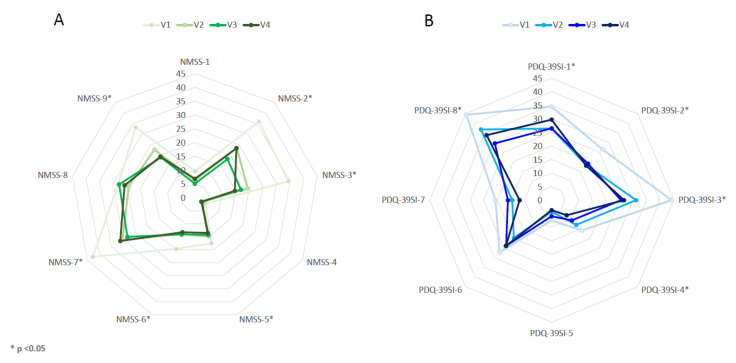
(**A**) Mean score on each domain of the NMSS scale at V1 (baseline), V2 (1 months ± 7 days), V3 (3 months ± 15 days) and V4 (6 months ± 15 days). The difference between V1 and V4 was significant for NMSS-2 (Sleep/fatigue) (*p* = 0.003), NMSS-3 (Depression/apathy) (*p* < 0.0001), NMSS-5 (Attention/memory) (*p* = 0.026), NMSS-6 (Gastrointestinal symptoms) (*p* = 0.010), NMSS-7 (Urinary symptoms) (*p* = 0.003) and NMSS-9 (Miscellaneous) (*p* < 0.0001). (**B**) Mean score on each domain of the PDQ-39SI at V1, V2, V3 and V4. The difference between V1 and V4 was significant for PDQ-39SI-1 (Mobility) (*p* = 0.037), PDQ-39SI-2, (Activities of daily living) (*p* = 0.014), PDQ-39SI-3 (Emotional well-being) (*p* < 0.0001), PDQ-39SI-4 (Stigmatization) (*p* = 0.021) and PDQ-39SI-8 (Pain and discomfort) (*p* = 0.018). *p* values were computed using the Wilcoxon signed-rank test. NMSs, Non-motor symptoms; PD, Parkinson’s disease; PDQ-39SI, 39-item Parkinson’s Disease Quality of Life Questionnaire Summary Index.

**Figure 3 brainsci-11-00316-f003:**
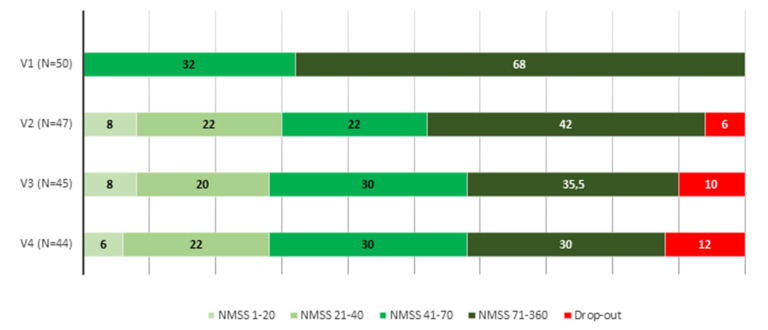
NMS burden (% of cases) with regards to the NMSS total score (0–20, slight burden; 21–40, moderate burden; 41–70, severe burden; 71–360 very severe burden) at V1, (baseline), V2 (1 months ± 7 days), V3 (3 months ± 15 days) and V4 (6 months ± 15 days).

**Table 1 brainsci-11-00316-t001:** Data about sociodemographic aspects, comorbidities, antiparkinsonian drugs and other therapies at baseline.

	*N*			*N*	
Age	50	68.50 ± 9.12	Time from diagnosis of PD	48	6.39 ± 5.06
Gender (females) (%)	50	58			
Race (%)	50		Motor fluctuations (%)	50	78
- Caucasian	50	98	Dyskinesia (%)	50	30
- Indian	50	2	Cognitive impairment (%)	50	4
Civil status (%):	27		Treatment for PD (%):	50	
- Married		70.4	- Levodopa		94
- Widowed		22.2	- Dopamine agonists:		66
- Other		7.4	* Pramipexole		36
			* Ropinirole		14
Living style (%)	29		* Rotigotine		18
- With the partner		65.5	- COMT inhibitor		34
- With another family member		27.6	* Entacapone		16
- Other		7.2	* Opicapone		18
			- Amantadine		8
Habitat (%):	30		- Anticholinergic drug		0
- Rural (<5.000)		13.4	L-dopa daily dose (mg)		612.23 ± 423.4
- Semiurban (5.000–20.000)		23.3	DA daily dose (mg)		234.33 ± 96.19
- Urban (>20.000)		63.3	LEDD (mg)		810.21 ± 518.14
Comorbidities (%):	50		Other treatments (%):	50	
- Arterial hypertension		40	- Antidepressant		32
- Diabetes mellitus		20	- Benzodiazepine		38
- Dyslipemia		36	- Antipsychotic		4
- Cardiopathy		10	- Analgesic		22
- Cardiac arrhythmia		12			
- Smoking		10	Number of anti-PD drugs	50	2.27 ± 0.94
- Alcohol consumption		12	Total number of drugs	50	6.36 ± 3.14

The results represent % or mean ± SD. Cognitive impairment is with regards to the opinion of the neurologist (no dementia; MMS ≥ 26).

**Table 2 brainsci-11-00316-t002:** Change in the score of the non-motor symptoms scale (NMSS) and other scales of the study from V1 (baseline) to V4 (6 months ± 15 days).

	V1	*N* (V1)	V4	*N* (V4)	∆ V1−V4	*p*
**MOTOR ASSESSMENT**						
**H&Y-OFF**	2.5 [2,3]	46	N. A.			
**H&Y-ON**	2 [2,2.5]	49	N. A.			
**UPDRS-III-ON**	24.63 ± 9.12	48	20.21 ± 9.81	39	−17.9%	**0.009**
**UPDRS-IV**	3.82 ± 2.55	50	2.82 ± 2.38	34	−25.4%	0.188
**FOGQ**	6.10 ± 5.23	48	5.68 ± 4.96	44	−6.9%	0.24
**NON MOTOR ASSESSMENT**						
**NMSS total score**	97.48 ± 43.70	50	59.91 ± 35.49	44	−38.5%	**<0.0001**
- Cardiovascular	9.58 ± 2.46	50	6.72 ± 11.94	44	−29.9%	0.268
- Sleep/fatigue	36.08 ± 21.77	50	23.15 ± 18.12	44	−35.8%	**0.002**
- Mood/apathy	34.42 ± 29.89	50	14.49 ± 19.63	44	−57.9%	**<0.0001**
- Perceptual symptoms	4.33 ± 8.67	50	2.84 ± 5.88	44	−34.4%	0.63
- Attention/memory	17.50 ± 17.09	50	13.32 ± 18.19	44	−23.9%	**0.026**
- Gastrointestinal symptoms	19.61 ± 18.01	50	13.13 ± 13.39	44	−33.0%	**0.01**
- Urinary symptos	42.72 ± 30.41	50	30.62 ± 23.94	44	−28.3%	**0.003**
- Sexual dysfunction	28.25 ± 35.69	50	25.28 ± 33.58	44	−10.5%	0.784
- Miscellaneous	33.33 ± 20.73	50	18.99 ± 14.03	44	−43.0%	**<0.0001**
**ESS**	9.20 ± 5.64	49	6.93 ± 5.11	44	−24.7%	**0.012**
**PSQI**	10.43 ± 4.02	47	8.36 ± 4.41	42	−19.8%	**0.001**
**BDI-II**	15.88 ± 10.46	50	10.18 ± 6.76	44	−35.9%	**<0.0001**
**KPPS**	40.04 ± 36.18	48	22.60 ± 21.42	44	−43.6%	**<0.0001**
- Musculoskeletal pain	48.44 ± 39.74	48	31.06 ± 30.15	44	−35.9%	**0.009**
- Chronic pain	11.89 ± 21.14	48	8.24 ± 15.24	44	−30.7%	0.636
- Fluctuation-related pain	11.11 ± 15.08	48	5.37 ± 10.74	44	−51.7%	**0.02**
- Nocturnal pain	23.18 ± 27.30	48	12.50 ± 26.17	44	−46.1%	**0.001**
- Oro-facial pain	2.49 ± 9.58	48	0.82 ± 3.20	44	−67.1%	1
- Discoloration, edema/swelling	11.46 ± 17.99	48	5.68 ± 11.65	44	50.40%	**0.009**
- Radicular pain	16.32 ± 30.60	48	9.66 ± 22.15	44	−40.1%	**0.048**
**VAS—PAIN**	4.61 ± 3.22	49	3.67 ± 2.69	43	−20.4%	0.071
**VAFS—Physical**	4.18 ± 2.84	49	3.64 ± 2.55	44	−12.9%	0.293
**VAFS—Mental**	3.14 ± 2.65	49	2.45 ± 2.79	44	−21.9%	0.118
**QOL AND AUTONOMY**						
**PDQ-39SI**	30.07 ± 17.61	49	21.24 ± 13.48	44	−29.4%	**<0.0001**
- Mobility	34.55 ± 27.79	49	29.09 ± 26.85	44	−15.8%	**0.037**
- Activities of daily living	26.50 ± 23.94	49	17.80 ± 17.96	44	−32.8%	**0.014**
- Emotional well-being	44.30 ± 29.34	49	26.33 ± 23.01	44	−40.6%	**<0.0001**
- Stigmatization	15.82 ± 22.79	49	7.67 ± 13.13	44	−51.5%	**0.021**
- Social support	7.48 ± 16.51	49	3.59 ± 12.63	44	−52.0%	0.302
- Cognition	27.17 ± 22.00	49	23.72 ± 22.49	44	−12.7%	0.876
- Communication	20.07 ± 26.73	49	12.12 ± 15.19	44	−39.6%	0.203
- Pain and discomfort	44.56 ± 27.35	49	33.33 ± 19.93	44	−25.2%	**0.018**
**ADLS**	81.40 ± 11.78	50	80.91 ± 16.39	44	−0.6%	0.845

*p* values were computed using the Wilcoxon signed-rank test. The results represent mean ± SD or median [p25, p75]. Domains of the NMSS and KPPS were expressed as a percentage to be able to establish comparisons on their severity between them. ADLS, Schwab and England Activities of Daily Living Scale; BDI, Beck Depression Inventory; ESS, Epworth Sleepiness Scale; FOGQ, Freezing Of Gait Questionnaire, H&Y: Hoehn & Yahr; KPPS, King’s PD Pain Scale; NMSS, Non-Motor Symptoms Scale; NPI, Neuropsychiatric Inventory; PDQ-39SI, 39-item Parkinson’s Disease Quality of Life Questionnaire Summary Index; PSQI, Pittsburgh Sleep Quality Index; UPDRS, Unified Parkinson’s Disease Rating Scale; VAFS, Visual Analog Fatigue Scale; VAS—Pain, Visual Analog Scale—Pain.

**Table 3 brainsci-11-00316-t003:** Adverse events in patients from V1 to V4.

Adverse Events	*N*
Total AEs, *N*	21
- Nausea	3
- Dyskenia	3
- Diurnal somnolence	2
- Urinary infection	1
- Dizziness	1
- Pneumonia	1
- Respiratory insufficiency	1
- Rash	1
- Diurnal somnolence	1
- Dry mouth	1
- Insomnia	1
- Increased appetite	1
- Abdominal pain	1
- Constipation	1
- Urinary incontinence	1
- Deep brain stimulation surgery	1
Patients with at least one AE, *N* (%)	16 (32)
At least possibly related AEs, *N*	14
Patients with at least possibly * related to safinamide AEs, *N* (%)	9 (18)
Total SAEs, *N*	5
- Urinary infection	1
- Pneumonia	1
- Respiratoy insufficiency	1
- Dizziness	1
- Deep brain stimulation surgery	1
Patients with at least one SAE, *N* (%)	5 (10)
At least possibly * related to safinamide SAEs, *N*	0
Patients with at least possibly being related to safinamide SAEs, *N* (%)	0 (0)
Patients with at least one AE leading to discontinuation, *N* (%)	4 (8)
Patients with at least one possibly * being related to safinamide AE leading to discontinuation *N* (%)	1 (2)
Deaths, *N* (%)	0 (0)

* Considered “possibly”, “probably” or “definitely” related to treatment (safinamide). AE, adverse event; SAE, serious adverse event.

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
