# Peer review of "Safinamide Improves Non-Motor Symptoms Burden in Parkinson’s Disease: An Open-Label Prospective Study"

_brainsci, 2021, doi:10.3390/brainsci11030316_

Round 1

Reviewer 1 Report

Santos-Garcia et al. provide data from the SAFINONMOTOR trial, which is an open-label, multi-site, single-country, observational, prospective study. They studied the effect of Safinamide (SAF) on non-motor symptoms (NMS) and quality of life (QoL) in 50 PD patients with a high burden of NMS, measured by various scales and questionnaires (which conform to international standards) over a time period of 6 months including 1 baseline and 3 follow-up visits. They find an overall improvement of the NMS burden and QoL, as well as improvements in specific sub-categories of NMS.

The data is of relevance, especially in the decision-making process during clinical routine in PD care, but also scientifically in deciphering the effect and its possibly underlying mechanisms of SAF. As a prospective, and -as far as I can oversee it- independent study, it adds to the increasing amount of knowledge about the drug in daily routine as compared to previous mostly post-hoc analyses of the approval studies and some smaller single-center trials.

Most of my questions that arose during reading the manuscript have already been answered in the discussion sections. Nevertheless, I have a few comments for the improvement of the manuscript:

  1. The authors conclude that “SAF improves NMS burden and QoL in PD patients at 6 months”. Even though in the context of the additional literature this might be correct, this conclusion can be drawn from the SAFINONMOTOR trial only for PD patients with a high NMS burden. PD patients with lower NMS burden have not been studied. This limitation has been adequately discussed in the section “discussion”, but the phrasing should be weakened in the abstract (ll. 19.20, ll. 30-31) and discussion (ll261-2, ll. 351-2).
  2. It seems to me that most of the results could possibly be explained by an hypothetical anti-depressant effect of SAF (most prominent changes were found in the subcategories pain, mood and sleep). This is somewhat surprising and expected at the same time. SAF is highly selective on MAO-B, so the “classical” anti-depressant effect of MAO-A-inhibition would not be the major mechanism. On the other hand, anti-depressant effects have been described in higher dosages of rasagiline, too (e.g. Korchounov et al. 2012, Clin Neuropharmacol). Same mechanism? Or is it due to the other mechanisms of SAF?  This has been discussed to a very small extend in the manuscript and I would be fine, if the authors wish to keep it that way. Nevertheless, I believe that the discussion would improve even further, if this was elaborated a bit more in depth.                                                   Additionally, one third of the participants took antidepressants at baseline, so I guess that most of those have the official diagnosis of a depressive disorder. Were those the ones that improved most?
  3. All the numbers in the tables have slipped. It took me quite a while to understand what number belongs to what item, which (honestly speaking) was a bit annoying. Please make sure, that the configuration of the tables is correct (I don’t believe that it is a problem of my viewer, I think it is a problem of the source document. Otherwise, I apologize for this comment).
  4. Since the NMSS is the main instrument in the study, for reasons of completeness, give a few more details in the methods section on how you assessed that scale. Was the interview conducted by a physician/study nurse, was it always the same interviewer (inter-rater-variability?) at all visits? Did the interviewer use the English version or did you use a Spanish version? If Spanish, was that translation validated previously?
  5. How did you assess AE/SAE? Did you go through a catalogue of symptoms or was it assessed with open questions?
  6. I did not exactly understand AEs/discontinuation due to dizziness. In ll. 249 you state that 2 AE dizziness led to withdrawal from the study, in ll. 250-1 you mention only one. In the table 3, there are two times dizziness (1 AE, 1 SAE). Please clarify. Furthermore you state in Table 3 “Patients with at least possibly* related to safinamide SAEs, N (%) 9 (18%)” -> so many SAEs (in text you say 5)?
  7. Figure 3: Due to the reduction of total numbers in time, the figure shows the data distorted. Either add a fifth item “drop-out” to the bars in V2-V4 or use absolute numbers (instead of relative) with the bars getting shorter with each visit.
  8. 53-4. Add your references no. 37,38 and 39 also here.

Some minor typos:

  1. Table 2: Hoehn (ll. 188)
  2. Table 3: Dyskinesia
  3. Table 3: Dizziness (twice)
  4. Figure 1: Why is there the explanation for the abbr. ICB?
  5. Figure 2: Disease (ll.229)
  6. 51 well tolerated
  7. 260 Discussion
  8. 430 Fahn

Author Response

Please, you can find attached the reply. Thanks!

Reviewer 2 Report

Dear Srs.

I carefully read this relevant study and I have some comments:

Comment 1: The Movement Disorder Society PD diagnostic criteria has become the gold standard in clinical practice and research settings (Postuma et al, Mov. Disord. 2015). The clinical studies should apply the MDS criteria instead of UK PD brain bank criteria.

Comment 2: The Unified Parkinson’s Disease Rating Scale (UPDRS) originally developed in the 1980s was reviewed in 2008 by Movement disorders society (MDS) and since then the recommendation is that the most current version is preferably used (MDS-UPDRS).

Comment 3: I did not find justification for the use of the Freezing of Gait Questionnaire.

Comment 4: lines 179/180 - Please revise this sentence; “…(bodily discomfort), 3 (emotional well-being), and 1 (mobility) in patients (Table 2). It seems that this sentences isn't finished.

Comment 5: For me, table 2 is deformed and it seems to me that a minus sign is missing in what appears to be the item Discoloration, edema/swelling” (-50.4%).

Comment 6: The reference number 5 seems incomplete.

Author Response

(The authors gave the same response as above.)
